# Sleep Disturbance Caused by Step Changes in Railway Noise Exposure and Earthquakes

**DOI:** 10.3390/ijerph21060783

**Published:** 2024-06-16

**Authors:** Takashi Morihara, Yasuhiro Murakami, Koji Shimoyama, Makoto Morinaga, Shigenori Yokoshima, Sohei Tsujimura, Yasuhiro Hiraguri, Takashi Yano

**Affiliations:** 1Department of Architecture, National Institute of Technology, Ishikawa College, Tsubata Town, Kahoku-gun, Ishikawa 929-0392, Japan; 2Sojo University, Kumamoto 860-0082, Japan; yasuhiro@arch.sojo-u.ac.jp; 3Aviation Environment Research Center, Organization of Airport Facilitation, Minato-ku, Tokyo 105-0011, Japan; k-shimoyama@aeif.or.jp; 4Department of Architecture, Daido University, Nagoya 457-8530, Japan; morinaga@daido-it.ac.jp; 5Research Division, Kanagawa Environmental Research Center, Hiratsuka 254-0014, Japan; 6Graduate School of Science and Engineering, Ibaraki University, Hitachi 316-8511, Japan; sohei.tsujimura.fifty@vc.ibaraki.ac.jp; 7Department of Architecture, Kindai University, Higashi Osaka 577-8502, Japan; hiraguri@arch.kindai.ac.jp; 8Kumamoto University, Kumamoto 860-8555, Japan; yano@gpo.kumamoto-u.ac.jp

**Keywords:** Shinkansen railway, conventional railway, sleep disturbance, step change, earthquake

## Abstract

Kyushu Shinkansen and conventional railway lines run parallel in the areas 5 km north of Kumamoto Station (northern area) and 12 km south of the station (southern area). Following the operation of the Kyushu Shinkansen Line in 2011, the adjacent conventional railway line in the north was elevated, a new station was operated in the south, and large earthquakes struck the Kumamoto area from March to April 2016. Sleep disturbances were compared before and after the interventions and earthquakes based on noise source (Shinkansen and conventional railways), area (northern and southern), and house type (detached and apartment) through socio-acoustic surveys from 2011 to 2017. The Shinkansen railway caused significantly less sleep disturbances in detached houses in the north after compared to before the earthquakes, presumably due to more frequent closures of bedroom windows in northern detached houses following the earthquakes. The Shinkansen railway caused significantly more sleep disturbances in apartments in the south after compared to before the earthquakes, presumably because the Shinkansen slowed down immediately after the earthquakes and returned to normal speed during the survey, suddenly increasing the noise exposure. There was no significant difference in the other six cases investigated. Overall, the interventions may not have caused significant differences in sleep disturbances. This article expands on the congress paper by Morihara et al. presented in the “Community Response to Noise” session at the 52nd International Congress and Exhibition on Noise Control Engineering in Makuhari, Japan, organized by the International Institute of Noise Control Engineering.

## 1. Introduction

Super-express railway (Shinkansen) lines have been operational across Japan since 1964. The Kyushu Shinkansen Line (KSL) began service in 2011, followed by the construction and partial operation of the Hokkaido, Hokuriku, and Nagasaki Shinkansen lines. Numerous socio-acoustic studies on super-express railway noise have been performed in Japan [1,2,3], while such studies are rare in other countries. Notably, Lambert [4] conducted a survey on TGV noise in France, and Zhang and Ma [5] investigated community response to super-express railway noise in China.

Although these surveys aimed to examine the effects of steady-state noise exposure, noise levels often change abruptly when new lines are opened or closed, or when interventions like railway elevation and the introduction of new stations are implemented. In the late 2000s, when 42 studies on the effects of step changes in traffic noise exposure or interventions were reviewed, Brown and van Kamp explained the causes of the change effect (the effects of noise exposure change, in addition to those of steady-state noise exposure) [6] and provided evidence on change effects [7]. They classified the change effect into excess response and under response. An excess response indicates that the response increases or decreases compared to that under steady-state conditions when the noise exposure increases or decreases. The opposite trend is called under response. They found a clear excess response to annoyance in source interventions for road traffic noise [7]. Brown and van Kamp systematically reviewed 48 intervention studies from 1980 to 2014 for the WHO environmental noise guidelines [8]; 37 studies were related to road traffic noise, 8 to aircraft noise, and only 3 to railway noise.

The three railway noise studies are as follows. Lam and Au [9] studied annoyance due to a step change in noise exposure following the opening of a new railway extension in Hong Kong but did not show activity disturbances. Although railway noise increased slightly, the annoyance decreased. They pointed out the importance of a positive image of the railway through the media. Moehler et al. [10] investigated the effects of rail grinding on railway noise reduction by focusing on the annoyance. Because this was a pilot study with 81 samples, the analysis was performed to provide preliminary results rather than test hypotheses. Schreckenberg et al. [11] explored the effects of informing residents about rail grinding on annoyance and activity disturbances and showed that providing this information decreased annoyance and some disturbances more than among residents who did not receive the information.

The KSL and conventional railway line (CRL) run parallel in areas 5 km north of Kumamoto Station (northern area) and 12 km south of the station (southern area). Following the opening of the KSL in 2011, the adjacent CRL was elevated in the northern area, a new station was opened in the southern area, and large earthquakes struck the Kumamoto area from March to April 2016. Tetsuya et al. [12] found that high levels of annoyance remained largely unchanged, while moderate annoyance levels decreased significantly following the start of operations on the Kyushu Shinkansen Line (KSL) in 2011, despite an increase in noise exposure (under response). Murakami et al. [13] extended the Kyushu Shinkansen noise survey to compare annoyance levels before and after the elevation of the CRL, the opening of a new station in March 2016, and the major earthquakes in April 2016. Their study indicated that the relationship between noise exposure and annoyance was significantly higher post-interventions and earthquakes, even though noise exposure levels were lower (under response).

While these studies were related to annoyance, only a few were about activity or sleep disturbances. According to Brown and van Kamp [8], the change effect appeared in noise annoyance but not at the same level as annoyance in activity interferences. For example, Breugelmans et al. [14] conducted a panel survey from 2002 to 2005 on the effects of Schiphol Airport’s new runway opening in 2003 and found no overreaction to severe sleep disturbance. Kastka et al. [15] found that there was no excess response in activity interferences by the construction of noise barriers. Following an annoyance study by Nguyen et al. [16], Morinaga et al. [17] investigated the effects of step changes in aircraft noise exposure on activity (listening, rest, and sleep) disturbances near the Hanoi Noi Bai International Airport. Despite finding a change in activity disturbance, the effect size for sleep disturbance (awakening) was smaller than that for annoyance.

In this section, we reveal the lack of intervention studies for railway noise, particularly on activity disturbances. This study investigated whether sleep disturbances were affected by the change in railway noise exposure caused by the interventions and the earthquakes using the same datasets as Murakami et al. [13] and provided material for future reviews on the effects of the intervention on response to environmental noises.

The 52nd International Congress and Exhibition on Noise Control Engineering (Inter-Noise 2023) was organized by the International Institute of Noise Control Engineering and held from 20 to 23 August 2023, in Makuhari, Japan. The congress comprised 87 sessions, covering various fields of noise control engineering, and included around 900 papers. This article expands on the congress paper by Morihara et al. [18], which was presented in the session “Community Response to Noise” at Inter-Noise 2023.

## 2. Methods

### 2.1. Social Surveys

Figure 1 illustrates the survey area that includes both the conventional railway line (CRL) and Kyushu Shinkansen Line (KSL), which run closely parallel to each other from 5 km north (northern area) to 12 km south (southern area) of Kumamoto Station. The surveyed strip extends 150 m east and west of the railway. Figure 2 shows the stages of KSL construction and the elevation of the CRL in the northern area. Initially, the CRL was moved to a first temporary line (Figure 2a), then the elevated KSL was constructed (Figure 2b) and began operations (Figure 2c). The CRL was later shifted to a second temporary line (Figure 2d and Figure 3), enabling the construction (Figure 2e) and subsequent operation (Figure 2f) of the elevated CRL in the northern area. The CRL, featuring a 1.5 m noise barrier, is slightly lower in elevation compared to the KSL, which has a 2 or 3 m noise barrier (Figure 4). In the southern area, the CRL was elevated in 2010 starting from Kumamoto Station to approximately 650 m south of the newly opened station (Figure 5) in March 2016, continuing on the ground further south (Figure 6). Surveys spanned from 2011 to 2017 (Table 1). Surveys I, II, III, and IV were conducted in both the north and south (2011), north only (2012), north only (2016), and south only (2017), respectively. For subsequent analyses, only data from Survey I in the southern area were used.

Roughly 3500 houses in the northern area and 3800 in the southern area were evenly distributed across each survey. The survey questionnaires and sleep disturbance scales remained consistent throughout all surveys. Common questions addressed aspects such as housing and residential environment, annoyance due to environmental factors, disruptions caused by railway operations, attitudes towards transportation, and personal factors.

The surveys were referred to as the “Survey on Living Environment”. Questionnaires, along with request letters, were distributed to one individual per household, selected using the nearest birthday method [19] which has facilitated a well-balanced distribution of demographic variables. Respondents returned the completed questionnaires by mail. Sleep disturbance was assessed using a 5-point verbal scale (not at all, slightly, moderately, very, and extremely) as recommended by the International Commission on Biological Effects of Noise (ICBEN) for evaluating annoyance [20]. Since train operations were infrequent between midnight and 6 a.m., difficulty falling asleep was used as an indicator of sleep disturbance. Although strong response such as “at least very sleep disturbed (VSD)” has been used as a sleep disturbance index, the numbers of VSD responses were too small, ranging from 2 to 20, for multivariate analyses. Thus, “at least moderately sleep disturbed (MSD)” was used as the dependent variable; following that, Bodin et al. [21] and Gidlof-Gunnarsson et al. [22] took “at least moderately annoyed” on the same 5-point verbal scale as the annoyance index.

### 2.2. Noise Measurements

Twenty-four-hour noise measurements were conducted at reference points for all surveys. Slow, A-weighted sound pressure levels were recorded every 0.1 s for a full day using sound level meters (RION NL-21 and NL-22) equipped with all-weather windscreens, positioned 1.2 m above the ground and 12.5 m from the nearest railway (reference point). Additionally, concurrent short-term noise measurements were taken at 4 to 5 locations, including reference points within 100 m of the railway. 

The energy-averaged *L*_AE_ (single-event A-weighted sound exposure level) was calculated using the top 10 events from 20 noise events for both the conventional railway and Shinkansen passing trains. Horizontal noise reduction equations were developed through logarithmic regression analysis between the averaged *L*_AE_ at simultaneous noise measurement points and distances. Vertical propagation equations were constructed using cubic regression analysis. This was carried out between the correction factor, which represents the difference in *L*_AE_ between the specified floor and the ground floor, and the designated floor number. These measurements were conducted at three apartments situated within 10–150 m of the railway, as detailed in Table A1.

### 2.3. Data Analysis

Following the elevation of the CRL in the northern area and the opening of a new station in the southern area, the Kumamoto earthquakes took place. To examine the impact of noise reduction due to the elevation, new station operations, and earthquakes on sleep disturbance, a multiple logistic regression analysis was conducted. The dependent variable was the presence or absence of a significant sleep disturbance (MSD), while the independent variables included nighttime noise level (*L*_night_), sex, age, noise sensitivity, frequency of opening or closing bedroom windows, and the survey year.

Age was categorized into two groups (≤50s and ≥60s) to ensure equal representation and balance between the younger and older participants for valid statistical analysis. Noise sensitivity was assessed on a 5-point verbal scale: (1) not at all, (2) slightly, (3) moderately, (4) very, and (5) extremely. These were then categorized into two groups: “not sensitive” (1–3) and “sensitive” (4–5). The frequency of opening or closing windows was measured seasonally using a 4-point verbal scale: (1) seldom or not at all, (2) sometimes, (3) often, and (4) always. These were re-categorized into “close” (1–2) and “open” (3–4). Due to the surveys being conducted in different seasons, the frequency of window opening in the summer was used for the years 2011, 2012, and 2017, while autumn data were used for 2016. All statistical analyses were performed using JMP 11 software (SAS Institute Inc., Cary, NC, USA, 2013).

## 3. Results

### 3.1. Basic Data

Table 2 summarizes the demographic variables for each year (area) and house type. The number of respondents living in detached houses ranged from 143 to 279, while those living in apartments ranged from 75 to 190. The study had more female than male respondents. Older adults were more likely to reside in detached houses, whereas no clear age trend was observed for those living in apartments.

Table 3 details the number of trains passing during the day, evening, and night for Surveys II (2012) and III (2016) in the north and Surveys I (2011) and IV (2017) in the south. In the northern area, the total numbers of local, Shinkansen, and freight trains were 80, 130, and 10, respectively, while in the southern area, the numbers were 140, 130, and 10, respectively. Although the numbers of local and freight trains remained consistent before and after the elevation, the number of Shinkansen trains in the north decreased due to operational changes. In contrast, the numbers of local and Shinkansen trains increased during the day in the southern area.

Table 4 presents the means and standard deviations of noise exposure levels across each survey. Since the elevation was completed in the north and a new station began operating in the south in March 2016, the *L*_night_ levels of the conventional railway decreased from 2012 to 2016 in the north and from 2011 to 2017 in the south for detached houses. However, while the *L*_night_ levels of the Shinkansen railway decreased from 2011 to 2017 in the south, they increased from 2012 to 2016 in the north. This increase was due to the number of passing Shinkansen trains rising from 13 in 2012 to 16 in 2016 and the number of detached houses close to the Shinkansen railway (within 50 m) growing from 10 (7%) in 2012 to 65 (23%) in 2016 (Table 5). The independence of frequencies and distance was tested using Pearson’s χ^2^ test (χ^2^ = 18.32, *p* < 0.01), revealing a significant difference in the distribution of residents in detached houses in the north before and after the earthquakes.

Table 6 shows the frequencies of bedroom window opening and closing before and after the earthquakes. Following the earthquakes, there was an overall increase in the frequency of bedroom window closures. For detached houses in the north, the relative frequency of window closures rose from 65% in 2012 to 91% in 2016, an increase of 26%. Increases were also observed for apartments in the north (14%), and for both detached houses (11%) and apartments (12%) in the south. Pearson’s χ^2^ test was used to assess the independence of window closure frequencies before and after the earthquakes, based on area and house type. The results were the following: χ^2^ = 44.00 (*p* < 0.01) for detached houses in the north, χ^2^ = 6.45 (*p* = 0.01) for apartments in the north, χ^2^ = 8.47 (*p* < 0.01) for detached houses in the south, and χ^2^ = 3.24 (*p* = 0.07) for apartments in the south.

### 3.2. Exposure–Response Relationships

Figure 7, Figure 8, Figure 9 and Figure 10 illustrate the *L*_night_–%MSD relationships before and after the earthquakes, categorized by area, house type, and noise source. In detached houses in the north, sleep disturbances caused by Shinkansen noise were lower after the earthquakes than before (Figure 7b). Conversely, in apartments in the south, sleep disturbances caused by Shinkansen noise were more pronounced after the earthquakes than before (Figure 10b). In Figure 7a, Figure 8, Figure 9 and Figure 10a, the curves intersect, indicating no clear difference in the *L*_night_–%MSD relationship before and after the earthquakes in these graphs.

### 3.3. Multiple Logistic Regression Analysis

Multiple logistic regression analyses were conducted with MSD as the dependent variable and *L*_night_, sex, age, noise sensitivity, bedroom window operation, and survey year (pre- or post-interventions and earthquakes) as independent variables for both the northern and southern areas, detached and apartment houses, and noise sources (conventional trains and Shinkansen). Detailed outcomes are provided in Table A2, Table A3, Table A4, Table A5, Table A6, Table A7, Table A8 and Table A9 of the Appendix A, with summarized findings in Table 7 and Table 8. 

*L*_night_ of the CRL and KSL was significant at the 5% level, except for apartments in the southern area. Gender did not show significance in any instance. Age was found to have a significant impact on MSD only in relation to conventional railway noise and apartments in the northern area, indicating that individuals aged in their 50s and younger were more prone to sleep disturbance. Noise sensitivity significantly influenced MSD caused by both conventional railway and Shinkansen noise, except for apartments exposed to conventional railway noise in the south and both detached houses and apartments exposed to Shinkansen noise in the south. The frequency of bedroom window closure had a significant impact on MSD caused by Shinkansen noise for detached houses in the north, reflecting a trend of more residents in the northern area closing their windows at night compared to other scenarios. Interventions and earthquakes notably affected MSD caused by Shinkansen noise for detached houses in the north and apartment houses in the south, with no significant difference observed in the remaining six scenarios across survey years.

## 4. Discussion

The study examined the impacts of interventions (CRL elevation and new station operation) and earthquakes on sleep disturbances related to difficulty falling asleep across different areas, house types, and noise sources from 2011 to 2017. In six out of eight cases, there was no distinct variation in the *L*_night_–%MSD relationships before and after the interventions and earthquakes. However, notable differences were observed in the *L*_night_–%MSD relationships for the northern area, detached houses, and Shinkansen noise, as well as the southern area, apartments, and Shinkansen noise.

The average *L*_night_ of the Shinkansen railway saw a 5 dB increase from 2012 to 2016 for detached houses in the northern area (as shown in Table 4). Interestingly, despite this rise, sleep disturbances notably decreased (as indicated in Figure 7b through under response). This decrease in disturbances could possibly be attributed to more frequent window closures by detached house residents in the north compared to residents in other areas (as detailed in Table 6), leading to a reduction in indoor noise exposure. Given that the northern area is an older residential zone and situated closer to the epicenter than the southern area, the detached houses in the northern area experienced more severe damage. Residents in this area appeared to close their bedroom windows more frequently than those in the southern area. The level differences between the outside and inside for road traffic noise are estimated to be around 10 dB when windows are open and 25 dB when windows are closed in typical Japanese houses [23]. Since Shinkansen noise has almost the same frequency characteristics as road traffic noise [24], the level differences for window opening and closing may be similar to those of road traffic noise. As a result, indoor noise exposure likely decreased by more than 10 dB due to the windows being closed. 

Conversely, although the mean *L*_night_ of the Shinkansen railway slightly decreased from 2011 to 2017, sleep disturbances notably rose in the southern area, especially within apartments affected by Shinkansen railway noise. Following the earthquakes in the southern area, the Shinkansen’s maximum speed temporarily decreased to 70 km/h; however, it resumed to levels of 190–220 km/h in 2017. The corresponding power levels for Shinkansen speeds of 70 km/h and 200 km/h were estimated at approximately 100 dB and 114 dB respectively, calculated using a previously documented method [25]. Although this estimation method is applicable at speeds greater than 150 km/h, the measured values were consistent with those estimated at approximately 100 km/h [26]. The difference between the estimated and measured *L*_A,Smax_ from Shinkansen, with a speed of 103–116 km/h at points 25 m from the nearest railway, was negligible, from −0.64 to 0.18 dB. This indicates that noise exposure from the Shinkansen railway increased (approximately 14 dB) immediately after the earthquakes and through the survey period. Such unexpected sudden noise exposure changes may cause an increase in sleep disturbance. In the southern area, respondents in detached houses were exposed to Shinkansen noise diffracted by noise barriers, whereas those in apartments on the upper floors were directly exposed to Shinkansen noise. Thus, the effects of sudden changes in Shinkansen noise exposure may be greater for apartments than for detached houses in the south.

Sleep disturbances were compared before and after interventions and earthquakes based on the area, house type, and train type. The exposure–response relationships in the two cases differed significantly. The increased frequency of bedroom window closures in detached houses in the north after the earthquakes and the sudden increase in Shinkansen noise exposure from immediately after the earthquakes to the survey period explain the change effects. Unlike annoyance, step changes in noise exposure did not affect sleep disturbance, consistent with previous findings [8,18]. These findings are reasonable because noise exposure substantially affected activity disturbances, whereas annoyance is influenced by both noise and non-acoustic factors.

When examining the sleep disturbance profiles between conventional railway and Shinkansen noise, particularly for apartments in the northern area and detached houses in the southern area, based on Table 7 and Table 8, there seems to be no clear difference in the patterns. Only *L*_night_ consistently showed significance. This observation may further support the conclusion mentioned earlier.

This study had some limitations. This study is not longitudinal but cross-sectional since we did not engage the same participants for a series of surveys, specifically Surveys II and III in the northern area and Surveys I and IV in the southern area; thus, causal relationships are difficult to test. The surveys before the interventions were conducted in 2011 in the south and 2012 in the north, and those after the interventions were conducted in 2016 in the north and 2017 in the south because of personnel and financial constraints. Thus, social situations may have changed between 2011 and 2012 and between 2016 and 2017. Nguyen et al. [27] conducted four follow-up surveys at 3, 9, 35, and 44 months after a step change in aircraft noise exposure around Hanoi Noi Bai International Airport. They found no systematic differences in *L*_night_–%insomnia relationships among the four surveys. Such a difference in the survey date may not significantly affect the *L*_night_–%MSD relationship. Additionally, the 2016 survey season (autumn) differed from the others (summer); however, autumn in Kumamoto was quite warm. Because the response rates were low, and non-response surveys were not conducted, the respondent representativeness could not be confirmed. Murakami et al. [14] confirmed respondent homogeneity by comparing their responses to residential environments (green areas, townscapes, and views from houses). A non-response survey should be conducted in the future, considering the current situation in Japan. Sleep disturbances are usually caused by exposure to indoor noise. Measuring indoor noise exposure and asking for bedroom window directions is difficult because of privacy and security concerns. In Japan, the only method available is to estimate indoor noise exposure using representative sound insulation data.

## 5. Conclusions

Four surveys were conducted in the areas along the KSL and CRL before and after the interventions (CRL elevation and new station operation) and earthquakes in 2016. Sleep disturbances (difficulty falling asleep) were compared before (2011 and 2012) and after (2016 and 2017) the interventions, based on the area (north and south), noise source (Shinkansen and conventional railway), and house type (detached and apartment). Multiple logistic regression analysis showed that sleep disturbance caused by the Shinkansen railway for detached houses in the northern area was significantly lower after the earthquakes than before, while for apartments in the south, it was significantly higher after the earthquakes. These differences are explained by the more frequent closure of bedroom windows of detached houses in the north after the earthquakes. Moreover, the Shinkansen slowed down immediately after the earthquakes, returned to normal speed during the survey, and the noise exposure suddenly increased. Considering the lack of significant differences in the other six cases, the findings of this study suggest no significant difference in sleep disturbance due to the interventions. In this context, sleep disturbance may serve as a direct outcome for exposure–response relationships, as it seems to be impacted to a lesser extent by non-acoustic factors compared to annoyance and more significantly by noise exposure.

## Figures and Tables

**Figure 1 ijerph-21-00783-f001:**
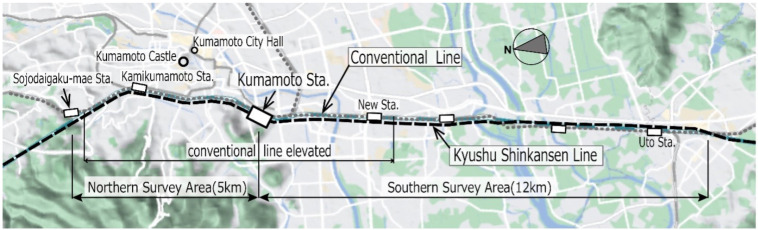
Survey area stretching from 5 km north of Kumamoto station to 12 km south of the station, with a width of 150 m on the east and west sides of the railways.

**Figure 2 ijerph-21-00783-f002:**
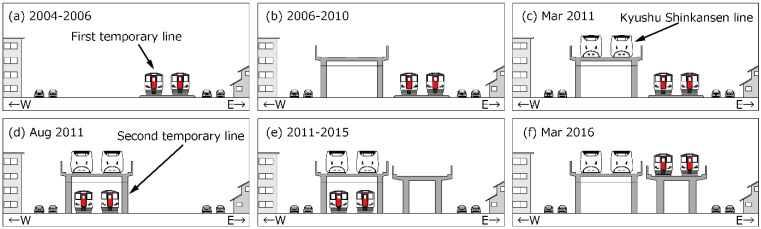
Process of railway construction in north area. (**a**) Conventional railway moved to the 1st temporary line; (**b**) construction of the elevated KSL; (**c**) operation of KSL; (**d**) conventional railway moved to the 2nd temporary line; (**e**) construction of the elevated conventional railway line; (**f**) operation of the elevated conventional railway line.

**Figure 3 ijerph-21-00783-f003:**
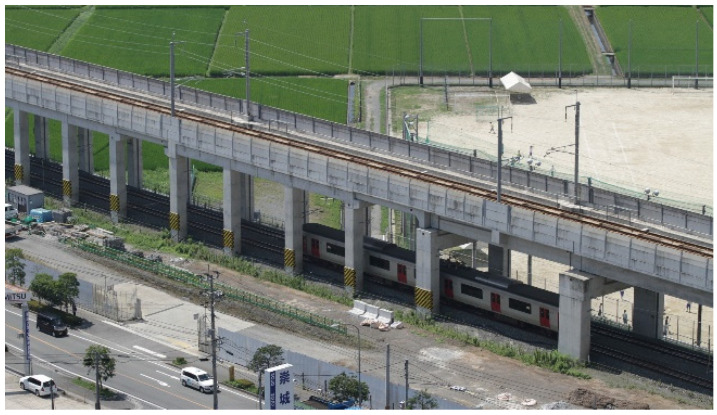
The 2nd temporary line in the northern area.

**Figure 4 ijerph-21-00783-f004:**
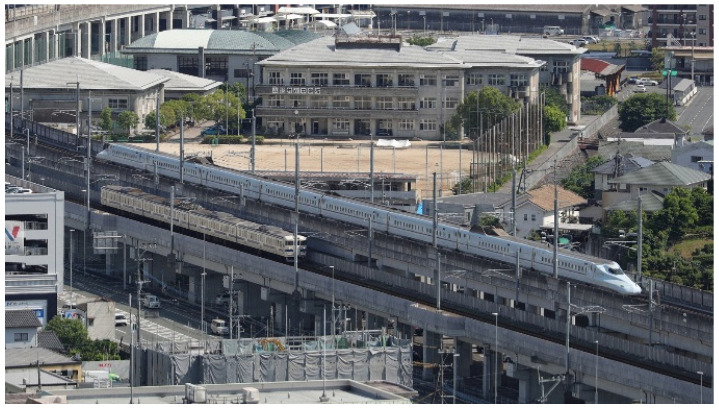
Elevated conventional railway and KSL in the northern area.

**Figure 5 ijerph-21-00783-f005:**
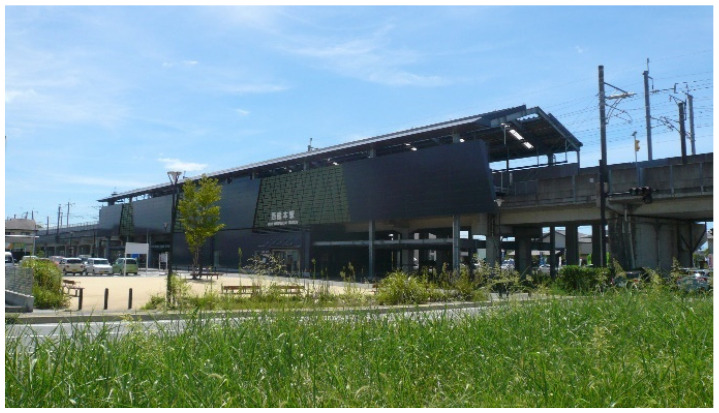
The new station in the southern area.

**Figure 6 ijerph-21-00783-f006:**
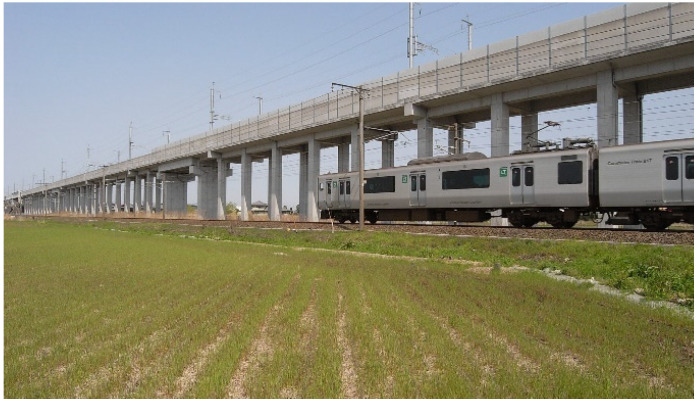
Conventional railway line on the ground and the elevated KSL in the southern area.

**Figure 7 ijerph-21-00783-f007:**
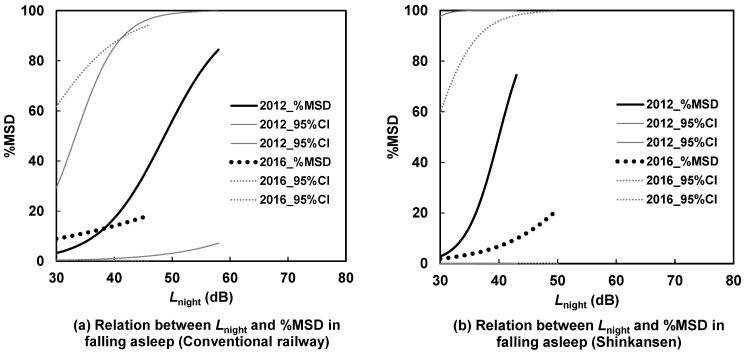
Contrast of *L*_night_–%MSD relationships for detached houses in the northern area between 2012 and 2016.

**Figure 8 ijerph-21-00783-f008:**
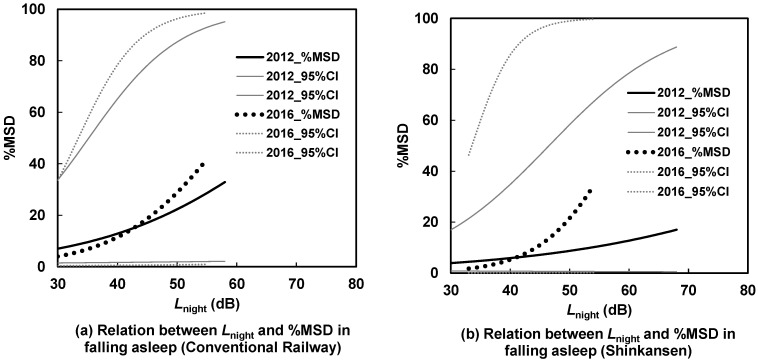
Contrast of *L*_night_–%MSD relationships for apartment houses in the northern area between 2012 and 2016.

**Figure 9 ijerph-21-00783-f009:**
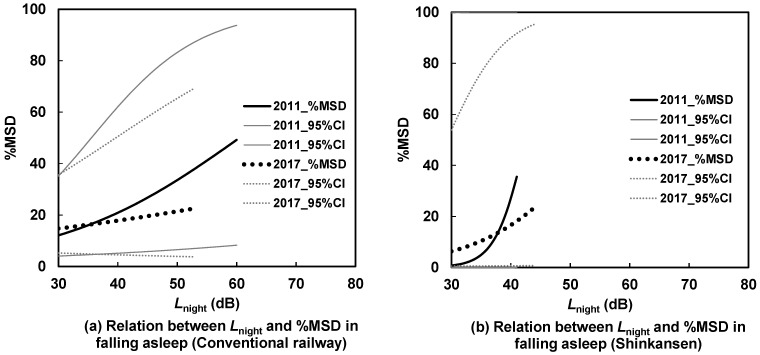
Contrast of *L*_night_–%MSD relationships for detached houses in the southern area between 2011 and 2017.

**Figure 10 ijerph-21-00783-f010:**
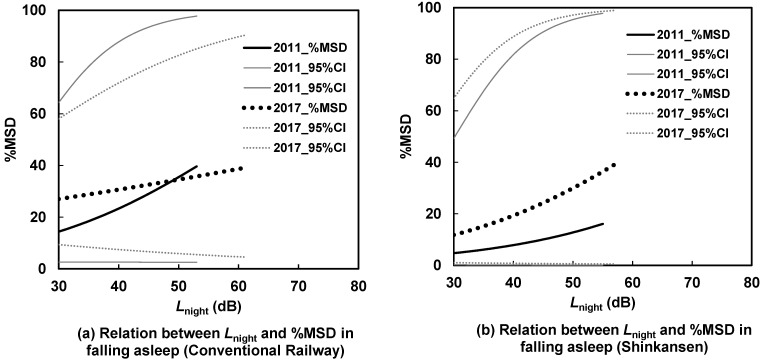
Contrast of *L*_night_–%MSD relationships for apartment houses in the southern area between 2011 and 2017.

**Table 1 ijerph-21-00783-t001:** Plan for social survey and earthquake incidence.

Date	Area	Railway Situation	Survey
12 March 2011		Opening of the Kyushu Shinkansen Line	
April–May 2011	North	Shinkansen + Conventional 1st temporary line	I
August–September 2011	South	Shinkansen + Conventional line	
July–August 2012	North	Shinkansen + Conventional 2nd temporary line	II
14–16 April 2016		Kumamoto earthquakes	
November–December 2016	North	Shinkansen + Conventional elevated line	III
July–September 2017	South	Shinkansen + Conventional line with new station	IV

**Table 2 ijerph-21-00783-t002:** Numbers of respondents, response rate, and the distribution of demographic variables.

Survey	II (2012)			III (2016)			I (2011)			IV (2017)		
House Type	D	A	Total	D	A	Total	D	A	Total	D	A	Total
No. of deliveries	312	787	1099	456	708	1164	612	710	1322	758	498	1256
No. of responses	143	190	333	279	120	399	236	139	375	253	75	328
Response rate (%)	46	24	30	61	17	34	39	20	28	33	15	26
Sex (%)												
Male	46	40	43	45	44	45	38	34	36	51	26	46
Female	54	60	57	55	56	55	62	66	64	49	74	54
Age (%)												
<30	4	29	19	1	16	5	5	11	7	3	9	5
30s	4	12	9	3	14	7	5	25	12	7	24	11
40s	14	16	15	9	10	10	15	21	17	14	19	15
50s	14	19	17	13	17	14	18	14	16	16	20	17
60s	29	12	19	32	26	30	26	17	23	23	15	21
≥70	36	12	22	41	17	34	32	12	24	37	12	32
No. of family members (%)												
1	24	52	40	23	37	27	18	22	19	17	23	19
2	30	18	23	37	36	36	34	29	32	34	40	35
3	24	13	18	21	13	19	18	26	21	21	21	21
4	15	15	15	10	11	10	15	17	16	15	12	14
5	5	2	33	5	3	5	9	4	7	9	4	8
6	2	1	1	3	-	2	4	1	3	3	-	2
7	-	-	-	1	-	1	1	-	1	1	-	1
8	-	-	-	-	-	-	1	-	1	-	-	-
Owned (%)	84	32	54	93	46	79	75	70	73	92	18	75

D: detached, A: apartment.

**Table 3 ijerph-21-00783-t003:** Number of passing trains during daytime, evening, and nighttime.

Survey	Period	Local		Shinkansen		Freight		Total	
		2012	2016	2012	2016	2012	2016	2012	2016
North area	Daytime (07:00–19:00)	56	56	98	87	4	2	158	145
	Evening (19:00–22:00)	13	12	24	24	3	3	40	39
	Nighttime (22:00–7:00)	15	14	13	16	5	7	33	37
	Total	84	82	135	127	12	12	231	221
Survey	Period	2011	2017	2011	2017	2011	2017	2011	2017
South area	Daytime (07:00–19:00)	94	101	71	84	4	2	169	187
	Evening (19:00–22:00)	24	24	21	17	2	3	47	44
	Nighttime (22:00–7:00)	23	22	30	29	6	5	59	56
	Total	141	147	122	130	12	10	275	287

**Table 4 ijerph-21-00783-t004:** Mean and standard deviation of noise exposures.

Area			North				South			
Year			2012		2016		2011		2017	
House Type	Source	Metric	Mean	S.D.	Mean	S.D.	Mean	S.D.	Mean	S.D.
Detached	Conventional railway	*L* _day_	39	9.0	37	4.0	42	9.5	42	7.5
		*L* _night_	34	8.9	32	4.0	40	9.5	36	8.8
		*L* _Aeq,24h_	37	9.0	36	4.1	42	9.5	41	7.8
		*L* _den_	42	9.0	40	4.1	47	9.5	44	8.3
	Shinkansen	*L* _day_	42	5.6	42	3.2	40	3.9	37	5.0
		*L* _night_	31	5.5	36	3.1	36	3.9	34	5.4
		*L* _Aeq,24h_	40	5.6	40	3.2	39	3.9	36	5.1
		*L* _den_	42	5.5	44	3.2	44	3.9	41	5.2
Apartment	Conventional railway	*L* _day_	49	8.2	45	6.4	43	7.9	46	10.4
		*L* _night_	45	8.3	40	6.4	41	7.9	40	11.8
		*L* _Aeq,24h_	48	8.2	44	6.4	43	7.9	45	10.7
		*L* _den_	53	8.2	48	6.4	48	7.9	49	11.2
	Shinkansen	*L* _day_	49	10.9	47	5.8	43	6.3	41	7.0
		*L* _night_	38	10.8	41	5.8	39	6.3	37	7.1
		*L* _Aeq,24h_	47	10.9	46	5.8	42	6.3	40	7.0
		*L* _den_	49	10.8	50	5.8	47	6.3	45	7.1

**Table 5 ijerph-21-00783-t005:** Number of detached house residents in the northern area and distance from Shinkansen railway.

	Distance from Shinkansen Railway *l* (m)			
Year	*l* ≤ 50	50 < *l* ≤ 100	*l* > 100	Total
2012	10	54	79	143
2016	65	99	115	279

**Table 6 ijerph-21-00783-t006:** Frequency of bedroom window opening before and after the earthquakes (relative frequency: %).

Area	Northern				Southern			
House Type	Detached		Apartment		Detached		Apartment	
Year	2012	2016	2012	2016	2011	2017	2011	2017
Close	91 (65)	239 (91)	115 (62)	87 (76)	162 (70)	200 (81)	92 (66)	57 (78)
Open	50 (35)	23 (9)	70 (38)	27 (24)	71 (30)	47 (19)	47 (34)	16 (22)
Total	141	262	185	114	233	247	139	73

**Table 7 ijerph-21-00783-t007:** Overview of multiple logistic regression analyses on difficulty falling asleep caused by conventional railway noise.

Item	Northern		Southern	
	Detached	Apartment	Detached	Apartment
*L* _night_	*	*	*	-
Sex	-	-	-	-
Age	-	*	-	-
Sensitivity	*	*	*	-
Frequency of opening windows	-	-	-	-
Year (interventions)	-	-	-	-

*: Significant at *p*-value < 0.05. -: Not significant.

**Table 8 ijerph-21-00783-t008:** Overview of multiple logistic regression analyses on difficulty falling asleep caused by Shinkansen railway noise.

Item	Northern		Southern	
	Detached	Apartment	Detached	Apartment
*L* _night_	*	*	*	-
Sex	-	-	-	-
Age	-	-	-	-
Sensitivity	*	*	-	-
Frequency of opening windows	*	-	-	-
Year (interventions)	*	-	-	*

*: Significant at *p*-value < 0.05. -: Not significant.

## Data Availability

After screening the datasets, they will be uploaded to the Socio-Acoustic Survey Data Archive (SASDA) at: https://www.ince-j.or.jp/old/04/04_page/04_doc/bunkakai/shachodata/.

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
