# Peer review of "Sleep Disturbance Caused by Step Changes in Railway Noise Exposure and Earthquakes"

_ijerph, 2024, doi:10.3390/ijerph21060783_

Round 1

Reviewer 1 Report

Comments and Suggestions for Authors

I think this is a valuable research report that examines the impact of infrastructure changes and natural disasters. I don't have much comment on the content, but I do have a couple of small ones.

(1) Is it correct to understand the impact of the earthquake as follows?

- The impact of the earthquake observed in this study was only a change in the speed of the Shinkansen bullet train service, and there was no change in non-acoustic factors such as the lifestyle of residents or their attitude toward the environment.

- As a result, the dose-response relationship was not significantly affected.

(2) If it is possible to compare Shinkansen and conventional lines, it would be good to mention them in comparison with previous reports.

Comments on the Quality of English Language

Some of the English expressions were not very comprehensible and should be improved. For example, 

 L.339; for the nothern area, detached houses, and Shinkansen noise, and the ~

 L.410; as it seemed to be by non-acoustic factors not so much as ~

Author Response

The authors sincerely appreciate the reviewer for providing valuable comments. Each comment will be addressed individually, and the manuscript will be revised in red. The sections highlighted by the assistant editor for their resemblance to published papers have also been amended.

I think this is a valuable research report that examines the impact of infrastructure changes and natural disasters. I don't have much comment on the content, but I do have a couple of small ones.

(1) Is it correct to understand the impact of the earthquake as follows?

- The impact of the earthquake observed in this study was only a change in the speed of the Shinkansen bullet train service, and there was no change in non-acoustic factors such as the lifestyle of residents or their attitude toward the environment.

In respect to the southern area, that is accurate. However, residents in detached houses in the northern area showed a tendency to close their bedroom windows more frequently compared to others. As detached houses in the northern area experienced more significant impacts from the earthquakes, they seemed to exhibit greater sensitivity to Shinkansen noise.

- As a result, the dose-response relationship was not significantly affected.

 Regarding the interventions, such as the elevation of the conventional railway in the northern area and the operation of the new station in the southern area, we concur that the exposure-response relationship was not significantly impacted. Thank you for your comment.

(2) If it is possible to compare Shinkansen and conventional lines, it would be good to mention them in comparison with previous reports.

Thank you for your advice. While we have previously published several papers on the health effects of Shinkansen noise, the focus was on annoyance rather than sleep disturbance. Therefore, in the "Discussion" section of this study, we compared the sleep disturbance patterns as follows: When examining the sleep disturbance profiles between conventional railway and Shinkansen noise, particularly for apartments in the northern area and detached houses in the southern area, based on Tables 7 and 8, there seems to be no clear difference in the patterns. Only Lnight consistently showed significance. This observation may further support the conclusion mentioned earlier.

Comments on the Quality of English Language

Some of the English expressions were not very comprehensible and should be improved. For example, 

 L.339; for the nothern area, detached houses, and Shinkansen noise, and the ~

This part was revised in the first paragraph of Discussion as follows: However, notable differences were observed in the Lnight–%MSD relationships for the northern area, detached houses, and Shinkansen noise, as well as the southern area, apartments, and Shinkansen noise.

L.410; as it seemed to be by non-acoustic factors not so much as ~

This sentence is revised in Conclusions as follows: In this context, sleep disturbance may serve as a direct outcome for exposure-response relationships, as it seems to be impacted to a lesser extent by non-acoustic factors compared to annoyance, and more significantly by noise exposure.

Reviewer 2 Report

Comments and Suggestions for Authors

This survey work is generally well-written. One of my major concern, as mentioned by the authors, is the influence of indoor noise. To what extent current survey research can truly reflect the influence of step changes in railway noise exposure and earthquakes on sleep disturbance, please provide a more in-depth discussion. My other comments are:

Page 1, line 41, change “seldom” to “seldomly”.

Page 3, line 121. Provide a reference for the nearest building method if it is well-established. Otherwise, an explanation of the method should be given.

A more comprehensive information might be provided in the caption of Figure 1.

Page 5, line 165. What are the vertical propagation equations? Please specify.

Page 5, line 176. Any reason that 50s and 60s are used as the criteria to classify the testing group? Is any reference to support this classification?

Page 12, line 374. “This study is not longitudinal but cross-sectional”, I do not understand what do you mean by this statement.

Comments on the Quality of English Language

authors are suggested to thoroughly review the writing as some grammar errors are detected.

Author Response

The authors sincerely appreciate the reviewer for providing valuable comments. Each comment will be addressed individually, and the manuscript will be revised in red. The sections highlighted by the assistant editor for their resemblance to published papers have also been amended.

This survey work is generally well-written. One of my major concern, as mentioned by the authors, is the influence of indoor noise. To what extent current survey research can truly reflect the influence of step changes in railway noise exposure and earthquakes on sleep disturbance, please provide a more in-depth discussion.

Thank you for your comments. We included an explanation on the impact of indoor noise exposure in the Discussion section as follows: Given that the northern area is an older residential zone and situated closer to the epicenter than the southern area, the detached houses in the northern area experienced more severe damage. Residents in this area appeared to close their bedroom windows more frequently than those in the southern area. The level differences between the outside and inside for road traffic noise are estimated to be around 10 dB when windows are open and 25 dB when windows are closed in typical Japanese houses [22]. Since Shinkansen noise has almost the same frequency characteristics as road traffic noise [23], the level differences for window opening and closing may be similar to those of road traffic noise. As a result, indoor noise exposure likely decreased by more than 10 dB due to the windows being closed.

My other comments are:

Page 1, line 41, change “seldom” to “seldomly”.

This part is revised as follows: while such studies are rare in other countries.

Page 3, line 121. Provide a reference for the nearest building method if it is well-established. Otherwise, an explanation of the method should be given.

Regarding the nearest birthday method, although it may not be conclusive, it has facilitated a well-balanced distribution of demographic variables. A concise explanation of the method has been included in Section 2.1. One of our co-authors engaged in a cross-cultural study on community responses to road traffic noise in collaboration with researchers from the University of Gothenburg. Following this, comparable methods, such as the nearest birthday approach, were implemented in line with the university's researchers' practices. In a subsequent publication, the authors utilized the method outlined in Section II B, page 3462.

Pederson E, Persson Waye K. Perception and annoyance due to wind turbine noise—a dose–response relationship. J. Acoust. Soc. Am. 116 (6): 3460-3470

A more comprehensive information might be provided in the caption of Figure 1.

The revised caption is as follows: Survey area stretching from 5 km north of Kumamoto station to 12 km south of the station, with a width of 150 m on the east and west sides of the railways.

Page 5, line 165. What are the vertical propagation equations? Please specify.

Thank you for your input. We discovered an error where vertical noise measurements were carried out at three apartments instead of five. The revised section reads as follows: Vertical propagation equations were constructed using cubic regression analysis. This was carried out between the correction factor, which represents the difference in LAE (single-event A-weighted sound exposure level) between the specified floor and the ground floor, and the designated floor number. These measurements were conducted at three apartments situated within 10–150 m of the railway, as detailed in Table A1.

Page 5, line 176. Any reason that ≤50s and ≥60s are used as the criteria to classify the testing group? Is any reference to support this classification?

In order to guarantee a balanced sample size and valid statistical analysis for both younger and older age groups, the participants were classified into distinct categories based on age. The clarification has been included in Section 2.3 as follows: Age was categorized into two groups (≤50s and ≥60s) to ensure equal representation and balance between the younger and older participants for valid statistical analysis.

Page 12, line 374. “This study is not longitudinal but cross-sectional”, I do not understand what do you mean by this statement.

We did not engage the same participants for a series of surveys, specifically Surveys II and III in the northern area, and Surveys I and IV in the southern area. Thus, this study is cross-sectional rather than longitudinal.

Comments on the Quality of English Language

authors are suggested to thoroughly review the writing as some grammar errors are detected.

Thank you for your advice. We have revised the manuscript further.